# Bioactive Natural Compounds with Antiplatelet and Anticoagulant Activity and Their Potential Role in the Treatment of Thrombotic Disorders

**DOI:** 10.3390/life11101095

**Published:** 2021-10-15

**Authors:** Stefania Lamponi

**Affiliations:** Department of Biotechnologies, Chemistry and Pharmacy and SienabioACTIVE, University of Siena, Via Aldo Moro 2, 53100 Siena, Italy; Stefania.Lamponi@unisi.it; Tel.: +39-0577-234386; Fax: +39-0577-234254

**Keywords:** bioactive natural compounds, anticoagulant, antiplatelet, hemostasis, thrombotic disorders

## Abstract

Natural anticoagulant drugs can be obtained from plants, rich in secondary bioactive metabolites which, in addition to being effective antioxidants, also possess anticoagulant and antiplatelet properties and, for this reason, can be excellent candidates for the treatment of thrombotic diseases. This review reports an overview of the hemostatic process and thrombotic disorders together with data on plants, more and less common from around the world, containing bioactive compounds characterized by antiplatelet and anticoagulant activity. The reported literature was obtained from Medline, PubMed, Elsevier, Web of Science, Google Scholar considering only articles in the English language, published in peer-reviewed journals. The number of citations of the articles and the impact factor of the journals were other parameters used to select the scientific papers to be included in the review. The analysis of the literature data selected demonstrates that many plants’ bioactive compounds show antiplatelet and anticoagulant activity that make them potential candidates to be used as new natural compounds able to interfere with both primary and secondary hemostasis. Moreover, they could be used together with anticoagulants currently administered in clinical practice to increase their efficacy and to reduce complications in the treatment of thrombotic disorders.

## 1. Introduction

The fluidity of blood through blood vessels in physiological conditions and the formation of thrombi to prevent bleeding from injured vessels are phenomena which depend on a complex biological process defined as hemostasis [1]. The hemostatic process is regulated by a balance between procoagulant and anticoagulant factors. The alteration of this balance, because of genetic or acquired conditions, can induce a state of hypercoagulability culminating with an increased tendency to thrombi formation [2]. Factors able to induce hypercoagulability are those described by the Virchow, i.e., endothelial lesions of vascular wall, decreased blood flow rate, and alteration of blood composition [3,4]. Vessel wall trauma and inflammation can be the causes of endothelium injury [5]; patients’ immobilization can decrease blood flow rate; both genetic and acquired conditions are able to induce alteration of blood composition [6]. Among genetic conditions, the most important are the presence of Factor V Leiden [7], mutation of the prothrombin gene [8], deficiencies of antithrombin, protein C and protein S [9], high homocysteine, fibrinogen and plasminogen activator inhibitor concentration [10,11,12], and alteration of the fibrinolytic system [13]. Pathologies, such as cancer, which alter the hemostatic system by acting on mechanism controlling thrombin formation and activity inducing thrombotic events, are examples of acquired situations able to modify blood composition [14,15,16] inducing a hypercoagulability state whose pathological consequence can be Venous Thromboembolism (VTE). Both Deep Vein Thrombosis (DVT) and Pulmonary Embolism (PE) are VTE diseases, associated with a high morbidity and mortality worldwide [17,18]. The clinical treatment of VTE is usually performed by systemic anticoagulation [19] and in recent years, significant advances have been made in its treatment by subcutaneous administration of low-molecular-weight-heparin (LMWH) and vitamin K antagonists (VKAs) [20,21,22,23,24,25]. Anyway, the treatment with LMWH and VKAs, being associated with complications such as risk of bleeding, requires monitoring for the selection of optimal drug dosage [26,27].

To overcome the complications related to LMWH and VKAs administration, oral anticoagulants (DOACs) have recently been applied in clinical practice. They have the double advantage of a simple method of administration and of not requiring laboratory monitoring [28]. Several randomized clinical trials performed on cancer patients with associated VTE have demonstrated that oral anticoagulant treatment is able to reduce bleeding and potential interactions with chemotherapy drugs and may represent a good alternative to LMWH [28]. Moreover, oral anticoagulant drugs have been demonstrated to possess many advantages in comparison to VKAs in the prevention of stroke in non-valvular atrial fibrillation [29].

Despite the advantages of DOACs therapy in the treatment of thrombotic disorders in comparison to LMWH and VKAs therapies, it is important that the research of new anticoagulants drugs avoids complications for the patients, develops [30]. In this context, the wide therapeutic potential of bioactive natural compounds, with antiplatelet and anticoagulant activities, extracted from plants [31,32], should be expanded. Plants are, in fact, rich in bioactive secondary metabolites, potent antioxidants and anti-inflammatory compounds, also able to inhibit the activity of many enzymes, including plasma serine proteases, demonstrating anticoagulant and antiplatelet effects [33]. These natural compounds may be important alternative and/or complementary drugs for the treatment of hemostasis diseases, thanks to their natural origin, safety and low cost compared to synthetic drugs [34].

The purpose of this review is to provide an overview of hemostasis and thrombotic disorders together with a description of the main plants distributed in the world able to interfere with primary and secondary hemostasis and to report their active constituents and the possible mechanisms involved to evaluate their potential clinical application as novel sources of natural anticoagulant and antithrombotic drugs for the treatment of coagulation disorders.

## 2. Hemostasis

Hemostasis is the complex physiological process able to maintain the fluidity of blood through blood vessels in physiological conditions and to form thrombi to prevent bleeding from injured vessels. Hemostasis, commonly referred to as "blood coagulation", consists of two different sequential and interconnected mechanisms defined as primary and secondary hemostasis (Figure 1) [35].

Platelets and endothelium are the principal effectors of primary hemostasis, while secondary hemostasis corresponds to the activation of soluble plasma coagulation factors and culminated with the formation of the fibrin network [36].

Anyway, primary and secondary hemostasis are interconnected and share many factors (Figure 1).

### 2.1. Primary Hemostasis

Platelets or thrombocytes are small anucleate cells that originate from megakaryocytes in the bone marrow. They role in primary hemostasis is essential by adhering, activating, and aggregating at level of vascular injury after exposition to collagen and von Willebrand Factor (vWF), and forming the platelet plug able to prevent blood loss from injured vascular endothelium [37]. Platelet adhesion happens by receptor–ligand interactions, and it is followed by platelet activation which, thanks to intracellular signaling, induces the secretion of the contents of intracellular α-granules and dense granules (Figure 2) [38].

In particular, because of their activation, the permeability of the platelet membrane is modified, allowing the entry of calcium ions, the release of chemotactic substances as well as adenosine diphosphate (ADP), serotonin and thromboxane A2, and also recalling the plasma coagulation factors on their surface [39]. The release of factor V and acid phospholipids further contributes to the amplification of the process. At last, platelets aggregate forming the platelet plug which stop blood loss from the injured vascular endothelium [40]. 

Moreover, negatively charged phosphatidylserine present on platelet membrane induces localized coagulation by thrombin formation amplifying secondary hemostasis [41,42,43,44,45]. Thrombin is also a potent platelet activating agent thanks to its specific receptor located on the thrombocyte’s membrane, and it modulates its own generation by activating coagulation factors V, VIII and XI, contributing to the formation of the fibrin clot [46].

Besides their role in platelet plug formation, the platelet surface is the substrate where tenase and prothrombinase complexes, involved in secondary hemostasis, form [47,48,49].

Therefore, platelets play a central role in the hemostatic process, being actively involved not only in primary hemostasis but also in thrombin generation, which amplifies the blood coagulation cascade, i.e., secondary hemostasis [50,51,52].

### 2.2. Secondary Hemostasis

Secondary hemostasis involves the activation of plasmatic coagulation proteins, and it is regulated by a series of positive and negative feedback mechanisms. In particular, a series of reactions involving the activation of zymogens (inert precursors of enzymes) and protein cofactors in, respectively, active serine proteases and cofactors form the blood coagulation cascade [53].

Two main pathways activate the blood coagulation cascade, i.e., secondary hemostasis: the extrinsic or tissue factor pathway and the intrinsic or contact pathway which both converge in the common pathway culminating with the formation of a clot formed by covalent-stabilized fibrin network [54] (Figure 3).

#### 2.2.1. Extrinsic or Tissue Factor Pathway

The tissue factor pathway is thus defined by the integral cell-surface protein known as tissue factor (TF) which activates it [55]. The term "extrinsic pathway" is because its activation requires the plasma to contact an "extrinsic" factor, i.e., TF, to be activated. 

The extrinsic/tissue factor pathway is activated when the membrane-bound protein TF, exposed by cells as a consequence of vascular endothelial cell lesion, contacts plasma containing factor VII or VIIa (Figure 3). The binding of factor VIIa to TF forms the TF-VIIa complex which, once bound to the cell membrane, becomes a potent coagulation cascade activator.

The TF-VIIa activates both factor IX and factor X [54] which must assemble on suitable membrane surfaces together with their own protein cofactors (respectively, factor VIIIa or factor Va) to propagate the coagulation cascade which culminates with thrombin activation, the last serine protease in the clotting cascade, able to cleavage fibrinogen into fibrin monomers that, upon polymerization, form a fibrin clot [55].

#### 2.2.2. Intrinsic or Contact Pathway

The contact or intrinsic pathway of coagulation is initiated by autoactivation of factor XII (Hageman factor) after its binding to an artificial or negatively charged surface in a process that involves high-molecular-weight kininogen (HMWK) and prekallikrein (PK) [56,57] (Figure 3). Factor XIIa, induces the conversion of prekallikrein (PK) to kallikrein [58] which in turn activates factor XII [59], and the activation of factor XI. Moreover, cleavage of HMWK also occurs. All these reactions culminate in the activation of factor IX [60] (Figure 3).

#### 2.2.3. Tenase Complex

Factor IX, activated by either the intrinsic or extrinsic pathway, forms a complex together with factor VIIIa, calcium and phospholipids, able to activate factor X (Figure 3). In order to prevent its activation, the von Willebrand (vWF) factor in plasma is bound to factor VIII and after its binding to platelets adhered at the site of vascular injury, activation of factor VIII and dissociation of vWF happen [61]. Thrombin can also directly activate factor VIII by a process which cannot be blocked by vWF [62].

However, it seems that the contact pathway has an important role in clot formation in vitro, but does not contribute to the hemostatic process in vivo [63].

#### 2.2.4. Common Pathway

The final common pathway of secondary hemostasis culminates with thrombin generation and fibrin clot formation (Figure 3).

Factor V is activated in the presence of thrombin and anionic phospholipids [64] and this step is followed by the formation of the prothrombinase complex (factor Xa linked to factor Va, its non-enzymatic cofactor, calcium and the surface of a phospholipid membrane). The conversion of prothrombin into thrombin can be catalyzed also by factor Xa alone and accelerated both by adding factor Va and by binding the complex to the phospholipid surface of activated platelets [65].

The conversion of prothrombin into thrombin can be also induced by the tenase complex [66,67]. Finally, soluble fibrinogen is converted into a network of insoluble fibrin, which closes the vascular injury, stopping blood loss.

## 3. Thrombotic Vascular Diseases

Thrombotic vascular diseases nowadays are one of the most important causes of morbidity and mortality worldwide and are associated with many pathologies. They are characterized by an increased probability of thrombus formation due to a state of hypercoagulability [68]. The fluidity of blood through blood vessels in physiological conditions depends on the balance between procoagulant and anticoagulant factors. When this balance is altered, bleeding or thrombotic diseases can occur due to the modification of the mechanisms that control both the formation and activity of thrombin and platelet reactivity. [69]. In particular, the prevalence of procoagulant factors over the anticoagulant ones, determines a situation of hypercoagulability which induces VTE, a pathology that includes DVT and PE [70,71]. In DVT, clot formation in a deep vein occurs and it can partially or completely block blood flow. Usually, DVTs occur in the lower leg, thigh, or pelvis, even if they also can form in other parts of the body. Although DVT itself does not compromise the patient’s life, the blood clot can potentially detach from the vessel wall, move with the bloodstream, block lung blood vessels (PE), and partially or totally close a pulmonary artery [72]. 

Moreover, also arterial embolism can be a frequent complication of atrial fibrillation able to block an artery supplying blood to the brain inducing ischemic stroke [73].

DVT is correlated also to chronic venous insufficiency of the lower extremities [74,75]. This condition is due to blood stasis which, in turn, causes chronic leg swelling, increased pressure and leg ulcers called venous stasis ulcers [76].

Among the many diseases that can induce DVT, one of the most important is cancer [77]. Indeed, it has been shown that the relative risk of developing DVT is higher in patients with this disease [78,79], suggesting a correlation between thrombosis and inflammation in tumors. Pathophysiological mechanisms, such as activation of coagulation and the fibrinolytic system, as well as inflammation, are also related to cancer and thrombosis [80,81,82]. Chronic inflammation is not only associated to an increased risk of cancer [83,84], but also represents a powerful prothrombotic stimulus that induces a greater regulation of procoagulant factors accompanied by a decrease in anticoagulants and inhibition of the fibrinolytic system [85,86].

SARS-CoV-2 virus infection also induces a prothrombotic state characterized by hypercoagulation, venous thrombosis (VT), thrombocytopenia and disseminated intravascular coagulation (DIC) [87,88,89]. A high percentage of hospitalized COVID-19 patients shows in fact coagulopathies highlighted by the alteration of many coagulation parameters among which the most relevant are thrombocytopenia, decrease in fibrinogen concentration, increase in D-dimer concentration and the prolongation of Prothrombin Time (PT) [90,91,92,93]. This COVID-19-induced prothrombotic state appears to be due to the mechanism by which the virus infects cells [94,95] and occurs through three different mechanisms.

The first mechanism arises from the binding between angiotensin converting enzyme 2 (ACE-2) receptors on endothelial cells and the virus. This binding triggers an inflammatory reaction resulting in endothelial damage and altered release of cytokines [96] which, in turn, induces vasoconstriction, platelet and endothelial activation and release of proinflammatory cytokines [97].

The second mechanism involves neutrophil extracellular traps (NETs) which, by amplifying secondary hemostasis, increase the degree of thrombin activation [98,99]. The consequence of the hyperactivation of neutrophils, in addition to inducing the formation of immuno-thrombosis, can cause disseminated intravascular coagulation (DIC), which significantly compromises the microcirculation [100,101].

The third mechanism concerns the alteration of complement activation which shares multiple factors with secondary hemostasis. Complement and hemostasis, in addition to being interconnected and in balance with each other, are also involved in the pathophysiology of various diseases [102]. In this regard, numerous experimental data have been reported demonstrating the procoagulant effect of the complement system in COVID-19 patients. This procoagulant effect appears to be mediated by the mannose-associated serine protease-2 (MASP-2), capable of activating thrombin [102] and therefore the coagulation cascade. Furthermore, complement proteins are able to indirectly alter the vascular endothelium and trigger secondary hemostasis, and to stimulate platelet aggregation and therefore primary hemostasis [103].

Thromboembolic complications can also occur frequently during pregnancy, the onset of which is strictly related to maternal age, idiopathic or secondary thrombosis in the patient’s history, type of birth, bed rest, obesity and thrombophilic diseases [104].

## 4. Bioactive Compounds from Plants

Plants have been widely used for therapeutic or preventive purposes since ancient times, thanks to their medicinal properties [105,106]. Nowadays, there is a growing interest in the study of bioactive molecules extracted from plant species, as they can be used for specific therapeutic purposes thanks to their specific activity [107]. The bioactivity of plant extracts and their potential application depend essentially on the nature and quantity of the classes of chemical compounds present [108].

Natural molecules extracted from plants are classified into primary and secondary metabolites depending on their role in plant growth and development [109,110]. Primary metabolites are compounds that are essential for plant metabolism, i.e., growth, development, and reproduction. They are produced at high levels because the plant needs them constantly for its own development. Although they are essential compounds for plants, they still do not have pharmacological effects [110].

Primary metabolites are divided into two subgroups: primary essential metabolites and primary metabolic end products. Primary essential metabolites include proteins and carbohydrates, which form the structural and physiological basis of the plant. Primary metabolic end products, on the other hand, include compounds such as lactic acid and ethanol, which are the end products of various metabolic pathways.

Secondary metabolites are organic molecules that are not directly involved in plant growth, development, or reproduction and may be specific to different species [111]. Since secondary metabolites are not involved in the growth and development of the organism, the absence of these compounds has little to no effect on the survival of the plants, but they are important for other activities such as protection, competition, and interaction between species.

Secondary metabolites are classified into groups based on their biosynthetic origin and some of them are derived forms of primary metabolites. Bioactive molecules are essentially those secondary metabolites that have therapeutic, preventive, toxicological, immunostimulatory and anticoagulant activity [111]. A simple classification includes three main groups: terpenoids, phenolic compounds, and alkaloids [112]. Other groups include saponins, lipids, essential oils, carbohydrates, ketones, and others [113,114].

Phenolic compounds, mainly polyphenols and flavonoids, secondary metabolites found in large amounts in plants and in their extracts, are potent antioxidants with anti-inflammatory, antiplatelet [115] and anticoagulant activity [116,117]. Due to the described properties, polyphenols, flavonoids and polyphenol/flavonoid-rich extracts from plants could be very useful in both the prevention and treatment of thromboembolic complications. Polyphenols can be classified into many subclasses based on their bioactivity or chemical structure [118]. They are compounds with structural phenolic features that can be conjugated with various organic acids and carbohydrates. In plants, most of them are associated with sugars and are in the form of glycosides. They can be classified into different groups depending on the number of phenolic rings in their structure and the substituents attached to the rings. The main two groups are flavonoids and non-flavonoids [119]. The structure of flavonoids consists of two aromatic rings linked by three carbon atoms forming an oxygenated heterocycle. They are divided into six main subgroups as a function of the type of heterocycle: flavones, flavonols, flavanones, flavanonols, flavanols, anthocyanins, isoflavones, neoflavonoids, and chalcones.

Non-flavonoids are divided into the following subgroups: phenols, phenolic acids (hydroxybenzoic acid and hydroxycinnamic acid), benzoic aldehydes, tannins, acetophenones and phenylacetic acids, coumarins, benzophenones, xanthones, stilbenes, lignans and secoiridoids.

## 5. Antiplatelet and Anticoagulant Activity of Plants and Their Bioactive Compounds

The following are some plants whose extracts have been shown to have antiplatelet and anticoagulant activity. In addition, the antiplatelet and anticoagulant properties of the isolated compounds are also described to correlate the phytochemical composition of the extract with its ability to affect hemostasis.

### 5.1. Olea europaea (Oleaceae)

*Olea europaea* L. is an evergreen shrub or tree whose leaves are widely used as traditional remedies in European and Mediterranean countries. Extracts of olive fruits and leaves contain many potentially bioactive compounds that may have antioxidant, antihypertensive, antiatherogenic, anti-inflammatory, hypoglycemic, hypocholesterolemic, and antiplatelet properties [120,121]. One of these potentially bioactive constituents is the polyphenol oleuropein and other bioactive components are related secoiridoids, flavonoids, and triterpenes [122]. In particular, the presence of hydroxytyrosol (HT), a phenolic compound considered to be the most potent antioxidant after gallic acid [123], is associated with inhibition of platelet aggregation. Hydroxytyrosol acetate (HT-Ac), a polyphenol found in virgin olive oil, showed a stronger antiplatelet effect than HT and a similar effect to acetylsalicylic acid, as it can decrease platelet thromboxane synthesis in a concentration-dependent manner [124]. In particular, for a concentration of 400 μM of the phenolic compounds, HT reduce thromboxane production by 67 % and HT-Ac by 86 %. The synergistic effect of HT-Ac with HT has also been demonstrated, with more than twice the inhibition of platelet aggregation [125]. Several of the beneficial aspects of olive against cardiovascular disease through inhibition of platelet aggregation are associated with its constituent oleuropein [126]. Oleuropein and (+)-cycloolivil isolated from the wood of the olive tree reduced the ability of thrombin to stimulate platelet aggregation in a concentration range from 1 to 300 μM. Both compounds decreased thrombin-triggered Ca^2+^ release and entry into platelets to a similar extent as HT [127]. Polyphenols extracted from the leaves of olive *O. europaea* L. inhibited platelet activation and aggregation in vitro in healthy, non-smoking men, probably via their H_2_O_2_-scavenging properties, at a concentration of extract of 54.0 mg/mL [128]. Since polyphenols also exhibit synergistic behavior in mixed form, as found in olive leaf extracts high in oleuropein and other active polyphenols [129], the observed platelet inhibition may be due to a synergistic effect of different polyphenols, as opposed to oleuropein alone.

### 5.2. Chamomilla recutita L. (Asteracee)

*Chamomilla recutita* L. or *Matricaria recutita* L. Rauschert, or *Matricaria chamomilla*, is an annual herbaceous plant native to Europe and Western Asia. It is one of the most popular ingredients for herbal teas. Infusions and essential oils extracted from fresh or dried flower heads are traditionally used for medicinal purposes. The main constituents of the flowers are various phenolic compounds, especially the flavonoids apigenin, quercetin, patuletin, luteolin and their glucosides. Essential oil extracted from the flowers contain terpenoids, alphabisabolol and its oxides, and azulene, including chamazulene. Chamomile demonstrates to have moderate antioxidant and antimicrobial activity and significant platelet aggregation inhibition in vitro [130]. It is able to inhibit platelet aggregation induced by ADP and collagen and whole blood aggregation by collagen. Although unable to inhibit arachidonic acid- or thrombin-induced platelet aggregation, chamomile blocks thromboxane B2 synthesis induced by either ADP or collagen but does not significantly increase platelet cGMP levels [131]. Treatment of PRP with polysaccharide–polyphenol conjugates isolated from chamomile at concentrations of 100 µg/mL resulted in a dose-dependent decrease in platelet aggregation induced by ADP, arachidonic acid and collagen, to 88%, 78% and 71%, respectively. These results suggest that the inhibitory effect of conjugates on platelet aggregation occurs through several mechanisms. First, the clearest inhibitory effect on platelet aggregation by collagen suggests that conjugates may suppress platelet adhesion to endothelial cells and prevent the formation of atherosclerotic plaques. Second, the relatively weak inhibitory effect on platelet aggregation induced by ADP and arachidonic acid suggests that the conjugates might interact with platelet membrane receptors, inhibiting their binding ability with agonists. Finally, the conjugates could also inhibit platelet aggregation via the cyclooxygenase pathway, mimicking the effect of acetylsalicylic acid [132,133].

### 5.3. Allium sativum and Allium ursinum (Amaryllidaceae)

Garlic is a perennial flowering plant that grows from a bulb and is native to the Mediterranean region. It is a popular spice that has been used for both culinary and medicinal purposes for thousands of years [134]. Plants belonging to the Allium genus, such as garlic (*Allium sativum*), have been shown to exert multiple effects, including lowering blood pressure, reducing cholesterol and triglyceride levels, and promoting fibrinolysis and platelet aggregation inhibition [135,136,137,138]. Bioactive chemical constituents of garlic include sulfur compounds such as alliin, allicin, ajoene, allyl propyl disulfide, diallyl trisulfide (DATS), S-allylcysteine (SAC), vinyldithiins, S-allylmercaptocysteine; enzymes such as alliinase, peroxidases, tyrosinase; amino acids (arginine and others) and their glycosides; and some minerals. Fresh garlic, garlic powder, aged garlic and garlic oil have shown antiplatelet and anticoagulant effects by interfering with cyclooxygenase-mediated thromboxane synthesis thanks to 2-propenyl thiosulfate (2PTS) [139]. Platelet cyclooxygenase activity was inhibited by 2PTS in a dose-dependent manner up to 0.1 mM but tended to return to the control level at 1 mM. In contrast, the platelet reduced glutathione (GSH) concentration, decreased in a dose-dependent manner after treatment with 2PTS and a significant reduction was observed at 0.1 mM and 1 mM. Therefore, 2PTS-induced inhibition of platelet aggregation occurs as a result of inhibition of cyclooxygenase activity. Additionally, 2PTS may have a modulatory effect on platelet aggregation by affecting the platelet GSH concentration. Garlic compounds that contribute to antithrombotic activity include: alliin, ajoene, allicin, V-nyldithiins and DATS [140]. The results of an in vitro prospective study show that both extracts of *A. sativum* powder prepared from fresh *A. sativum* tubers and fresh *A. ursinum* (wild garlic) leaves obtained by maceration inhibit platelet aggregation induced via the ADP pathway in a dose-dependent manner and, to a lesser extent, aggregation induced by epinephrine, with similar efficacy. Inhibition of the ADP pathway by garlic extracts occurs with a mechanism of action comparable to that of the clinically used drug clopidogrel. The pharmacologically active component of the extracts appears to be lipophilic rather than hydrophilic, but the exact chemical is still unknown [141]. Previous studies suggest that organosulfur and phenolic compounds are partly responsible for the antiplatelet activity of Allium. Briggs et al. [142] investigated the in vitro platelet aggregation inhibitory effect of the following four thiosulfinates (TS): (ethyl methane-TS, MMTS; propyl propane-TS, PPTS; 2-propenyl 2-propene-TS, allicin; ethyl ethane-TS, EETS). All of them inhibited platelet aggregation, but with different magnitude. PPTS and allicin had the strongest antiplatelet activity at 0.4 mM, inhibiting aggregation by 90 and 89%, respectively. At the same concentration, EETS and MMTS were significantly weaker, inhibiting 74 and 26%, respectively. Moreover, combinations of different TS did not show additive effects on in vitro platelet aggregation inhibition, suggesting that the platelet inhibitory potential of Allium extracts cannot be predicted by the presence and treated dose of organosulfur components alone.

The results reported by Beretta et al. [143] suggest that organosulfur and phenolic compounds present in garlic extracts are involved to varying degrees in both platelet aggregation inhibition and antioxidant properties. While organosulfur and phenolic compounds contributed to a similar extent to in vitro platelet aggregation inhibition, phenolics were largely responsible for antioxidant activity. In particular, extracts of garlic inhibited platelet aggregation from 5 to 15–50 µL per mL of blood. The minimum platelet inhibitory doses of Allium extracts corresponded to 200–300 µL per mL of blood.

### 5.4. Rosmarinus officinalis (Lamiaceae)

*Rosmarinus officinalis* [144] is a Mediterranean evergreen shrub with aromatic needle-like leaves, now cultivated throughout the world. It is a rich source of antioxidants and anti-inflammatory compounds and can influence blood circulation. Recently, the ethanolic extract of *R. officinalis* has been shown to have an anticoagulant effect as it can inhibit thrombin activity by prolonging Thrombin Time (TT) for concentration values ranging from 2.4 × 10^−2^ to 9.6 × 10^−1^ mg of dry extract/mL [145]. This anticoagulant effect could be due to the triterpenes, ursolic acid and its isomer oleanolic acid, betulinic acid, carnosol and micromeric acid, present in the extracts, which are known for their anti-inflammatory activity [146,147,148]. Triterpenes and their derivatives are antioxidant and anti-inflammatory and possess anticoagulant activity by thrombin inactivation for concentration of 5–50 μg/mL [149]. Betulinic acid is able to decrease platelet aggregation induced by thrombin and inhibits antithrombin activity in a dose-dependent manner in rat model at a concentration of 10 mg/kg [150]. Carnosol has been shown to inhibit washed rabbit platelet aggregation induced by thrombin, collagen, arachidonic acid and U46619 in a dose-dependent manner in a concentration-dependent manner, with IC50 values of 39 ± 0.3, 34 ± 1.8, 29 ± 0.8 and 48 ± 2.9 μM [151].

### 5.5. Fragaria vesca, Echinacea purpurea and Erigeron canadensis L (Rosaceae and Asteraceae)

Plants from the Asteraceae family, commonly used in Polish folk medicine, have good antioxidant and anticoagulant activity. Macromolecular polysaccharides conjugated with polyphenols isolated from perennials of the Asteraceae family have high antioxidant activity and protect platelets from oxidative damage caused by biological oxidants [152]. Evaluation of the anticoagulant activity of extracts from different plants belonging to the Asteraceae and Rosaceae families showed that phytochemicals isolated from *Fragaria vesca* (Rosaceae) and *Echinacea purpurea* (Asteraceae) prolonged both Prothrombin Time (PT) and Activated Partial Thromboplastin Time (APTT) [153]. The results of the APTT test compared with the 5th International Standard of Unfractionated Heparin and expressed in international units (IU), demonstrated that the activity of the product isolated from *Fragaria vesca* corresponded to 2 IU/mg while the activity of the product isolated from *Echinacea purpurea* to 1.4 IU/mg. The observed anticoagulant effect could be due to the high content of hexuronic acids and phenolic glycoconjugates in the extracts. The polysaccharidic part, representing 32% of the total mass, contains mainly hexuronic acids and much smaller amounts of glucose, arabinose, galactose and some traces of mannose, xylose and rhamnose. The polyphenolic part is rich in hydroxyl residues and in free and esterified carboxyl groups [154]. Moreover, the anticoagulant effect of the extracts of *Erigeron canadensis* L. at the concentration as low as 390 μg/ml of standardized human blood plasma, and in PT test at the concentration of 1.56 mg/ml, seems to be due to the inhibitory action of the polyphenol–polysaccharide complexes isolated from the plant on antithrombin and heparin cofactor II [154]. 

### 5.6. Thymus atlanticus and Thymus zygis (Lamiaceae)

The genus Thymus [155] includes about 350 species of aromatic perennial herbaceous plants. *Thymus atlanticus* and *Thymus zygis* are the two species most commonly used in folk medicine for their anti-inflammatory properties. They are used in the form of infusions and decoctions to treat respiratory symptoms [156]. The aqueous extracts of *Thymus atlanticus* and *Thymus zygis*, rich in polyphenolic and flavonoid compounds such as caffeic acid, rosmarinic acid, quercetin, rutin, hyperoside, and luteolin-7-O-glucoside, show anticoagulant activity in vitro, as demonstrated by inhibition of blood clot formation in both APTT and TT assays in the highest tested concentration of all extract (5.70 μg in mixture) [157,158,159]. A study conducted with isolated compounds demonstrated that caffeic acid significantly inhibited thrombin-induced platelet aggregation at concentrations ranging from 100 to 500 μM, binding of fibrinogen to integrin αIIbβ3, and platelet-mediated clot retraction, and activated cAMP generation [160]. Rutin inhibits in vitro the platelet-activating factor responsible for free calcium concentration in platelets in a dose-dependent manner for concentrations of 68.3, 136, 274, 545 mumol/L [161]. In vitro and ex vivo coagulation studies show that APTT was significantly prolonged, and PT was also delayed by quercetin within doses of 10–30 μg [162]. In addition, bioinformatics analyses demonstrated that quercetin, together with procyanidin B2, cyanidin and silybin, had an inhibitory effect on FXa activity [163]. Bioinformatics analysis revealed that procyanidin B2, cyanidin, quercetin and silybin bind in the S1-S4 pockets near the fXa active site and block the access of substrates to Ser195. These data indicate that flavonoids could be potential structural bases for the development of new, safe, orally bioavailable direct factor Xa inhibitors in a natural way [163].

### 5.7. Licania rigida (Chrysobalanaceae)

The crude leaf extract (CELR) and ethyl acetate fraction (AFLR) of *Licania rigida* Benth showed anticoagulant activity in vitro by prolonging both APTT and PT and inhibiting the activity of factor Xa and IIa at a concentration of 50 and 100 mg/mL, respectively [164]. The anticoagulant effects of *L. rigida* extract may be due to the synergistic effects of polyphenols and their interactions with biomolecules that may affect the biological activity. Gallic acid, catechin, chlorogenic acid, caffeic acid, epicatechin, ellagic acid, rutin, quercitrin, quercetin, kaempferol, and kaempferol glycoside are the main components of *L. rigida* extracts that might be involved in their anticoagulant properties [165]. This implies that a plant extract may produce a more favorable response compared to the use of a single active ingredient [166].

### 5.8. Fumaria officinalis (Papaveracee)

*Fumaria officinalis* L. is a common plant in Tunisia with a high content of phenols and flavonoids and an anticoagulant effect in terms of prolongation of APTT and PT [167]. Among the extracts obtained with different solvents, the methanolic one showed the highest total phenolic and flavonoid content and the best antioxidant and anticoagulant properties [167].

### 5.9. Careya arborea (Lecythidaceae)

*Careya arborea* Roxb [168] is a deciduous tree in the family Lecythidaceae, native to South Asia. [169]. The methanolic extract of *C. arborea*, rich in phenolic compounds, has good antioxidant properties and an anticoagulant effect comparable to that of warfarin, a drug commonly used to treat thrombotic disorders such as DVP and PE [170]. The extract exerts its anticoagulant effect by increasing APTT, PT and TT [171] probably thanks to the high content of gallic acid, 3,4-dihydroxybenzoic acid, quercetin 3-O-glucopyranoside, kaempferol 3-O-glucopyranoside and quercetin 3-O-(6-O-glucopyranosyl)-gluco pyranoside [172].

### 5.10. Viola yedoensis (Violaceae)

*Viola yedoensis* Makino is widely used in traditional Chinese medicine as a remedy for Helicobacter pylori and HIV [173]. A new dicumarin named dimersculetin has been isolated from the ethyl acetate extract of the dried whole plants, along with another dicumarin, euphorbetin and esculetin [174]. Dimeresculetin is a 7-hydroxy-6-[(6,7-dihydroxy-2-oxo-2H-1-benzopyran-5-yl) oxy]-2H-1-benzopyran-2-one. Euphorbetine and esculetin showed anticoagulant activity with respect to APTT, PT and TT.

### 5.11. Euphorbia resinifera (Euforbiacee)

Plant latex is a source of anticoagulant compounds because it contains many types of proteases with ability to interfere with blood coagulation and antiplatelet activity but some of them show adverse effects [175]. EuRP-61 is a serine protease recently isolated from the plant latex of *Euphorbia resinifera* and, thanks to its anticoagulant activity, may be a new potential agent for the treatment of thrombosis [176]. This serine-protease can hydrolyze all chains of human fibrin clots, and it is not affected by human blood circulating inhibitors. EuRP-61 may influence secondary hemostasis by prolonging both PT and APTT in a concentration range from 0.5 to 4 µM. Moreover, the enzyme inhibits platelet aggregation via the ADP-receptor pathway and almost completely abolished the platelet function at concentrations ≥1 μM. Moreover, it is not toxic to human red blood cells in all the four Rh+ blood groups (A, B, O and AB) or human peripheral blood mononuclear cells (hPBMCs) [176]. 

### 5.12. Orobanche caryophyllacea, Phelipanche arenaria, and Phelipanche ramosa (Orobanchaceae)

*Orobanche caryophyllacea*, *Phelipanche arenaria*, and *Phelipanche ramose*, plants of the Orobanchaceae family, are rich in phenylpropanoid glycosides (PPGs) and possess a wide spectrum of activities, such as antimicrobial, anti-inflammatory, antioxidant, and anticoagulant ones [177]. Bioactivity evaluation of European broomrape (*O. caryophyllacea*, *P. arenaria*, *P. ramosa*) extracts and single isolated PPGs has demonstrated antioxidant and anticoagulant properties, in terms of prolongation of APTT, PT and TT, related to their chemical structure. In particular, the *O. caryophyllacea* extract, rich in acteoside, showed over 20% better antiradical potential than *P. ramosa* extract; the only one containing PPGs lacking a B-ring catechol moiety in the acyl unit. Only eight tested PPGs demonstrated antioxidant effect in human plasma treated with H_2_O_2_/Fe and three tested PPGs, tubuloside A, poliumoside and 3-O-methylpoliumoside, possessed both anticoagulant and antioxidant properties. Therefore, the anticoagulant potential of these compounds, as well as their antioxidant activity, seems to be related to their chemical structure, especially to the presence of acyl and catechol moieties [177].

### 5.13. Genipa americana (Rubiaceae)

*Genipa americana* is an exotic fruit largely consumed and well known, in Amazonian pharmacopeia, to treat anemia, measles and uterine cancer. It is also used as a diuretic, digestive, healing, laxative and antiseptic [178]. Two fractions, with different molecular weights, of glycoconjugates extracted from *Genipa americana* leaves (PE-Ga) composed mainly by arabinose, galactose and uronic acid, are able to prolong APTT and to inhibit by 48% the ADP-induced platelet aggregation at a concentration of 100 μg/μL [179]. Moreover, in vivo, these glycoconjugates inhibit venous thrombus formation and increase bleeding time at 1 mg/kg in Wistar rats [179].

Arabinogalactan–glycoconjugate fractions, named FI and FII demonstrated good antithrombotic activity in rats. FII prolonged the rat plasma coagulation time 5.5-fold in an APTT test and FI inhibited the platelet aggregation by 46%. Both FI and FII prevented thrombus formation and the bleeding time was not altered by any fractions [180].

### 5.14. Pseuderanthemum palatiferum (Acanthaceae)

*Pseuderanthemum palatiferum* is a plant first found in Northern Vietnam and which expanded throughout the country including the Mekong Delta region. The leaves are used in folk medicine of Vietnam and Thailand for treating various diseases such as hypertension, diarrhea, arthritis, hemorrhoids, stomachache, tumors, colitis, bleeding, wounds, constipation, flu, colon cancer, nephritis, and diabetes [181]. Polyphenolic–polysaccharide (PP) conjugates obtained from *Pseuderanthemum palatiferum* (Nees) Radlk leaves exhibit anticoagulant activity by prolonging both APTT and PT [182]. This effect is probably due to carbohydrate, phenolic, and protein constituents of the PP and to the seven different mono-sugars found in all the conjugates, i.e., arabinose, fucose, galactose, glucose, mannose, rhamnose, and xylose [182].

## 6. Conclusions

Many medicinal plants and their extracts contain anticoagulant and antiplatelet secondary metabolites that could be used as new plant-based hemostatic agents able to interfere with both primary and secondary hemostasis (Table 1). These natural compounds can be important complementary drugs for the recovery from hemostasis diseases due to their natural origin, safety, and low cost compared to synthetic drugs. In addition to these advantages, they can also be administered in association with anticoagulants currently used in clinical practice to increase the effectiveness of conventional anticoagulant therapy and to overcome its complications. Anyway, although there is no doubt that extracts obtained from plants could represent excellent candidates for the treatment of coagulation disorders related to many different diseases, their use shows some limits. One limit is represented by the concentration of the compound necessary to obtain functional consequences. For many cases, the concentration is too high, which, considering the pleiotropic effects of these compounds, argues against a potential clinical utility for the treatments. In addition to this, agronomical and processing conditions used in plant cultivation and extraction, respectively, may influence the chemical profile of the herbal preparations and, consequently, their pharmacological activities. Furthermore, the mechanisms of anticoagulant activity in the case of the vegetable preparation are the consequence of a synergistic effect of several compounds present in the extracts rather than of a single molecule. Finally, for most compounds and extracts, only the final effect are known, but not the underlying mechanism.

In order to overcome some of these limits, in vitro and in vivo studies aimed at identifying the compounds and their exact mechanism of action, as well as their adverse effects, are necessary. Once this is done, potential chemical modifications of natural compounds should be performed in order to improve their capacities such as half-life, specificity, IC50, etc. For example, introduction of OH groups at different positions on flavonoids either decreased or increased anti-thrombin potential [183]. Therefore, the study of the mechanism of activity of flavonoid-type compounds helps to design and develop more potent flavonoid-type inhibitors against thrombin and to improve the anticoagulant activity of natural compounds.

## Figures and Tables

**Figure 1 life-11-01095-f001:**
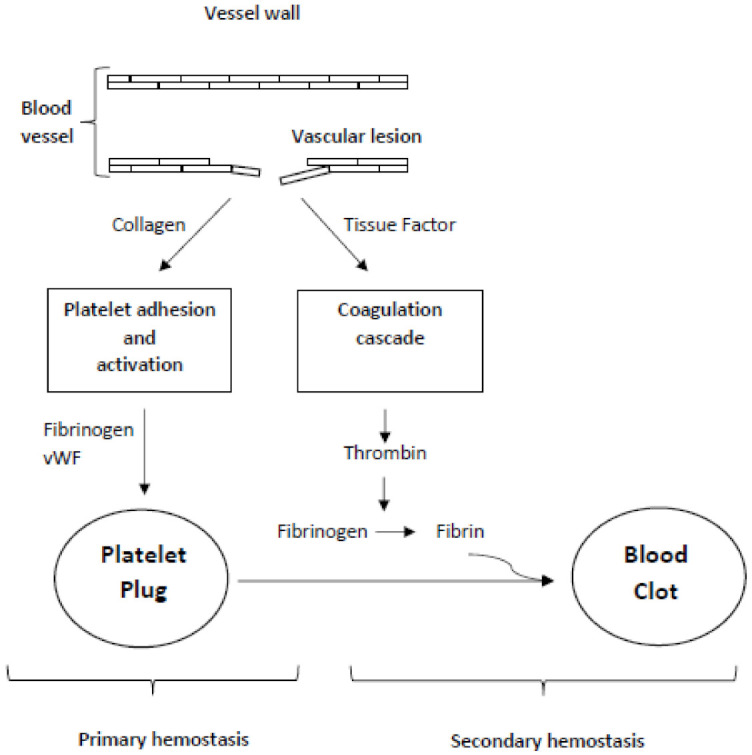
Interconnection of primary and secondary hemostasis.

**Figure 2 life-11-01095-f002:**
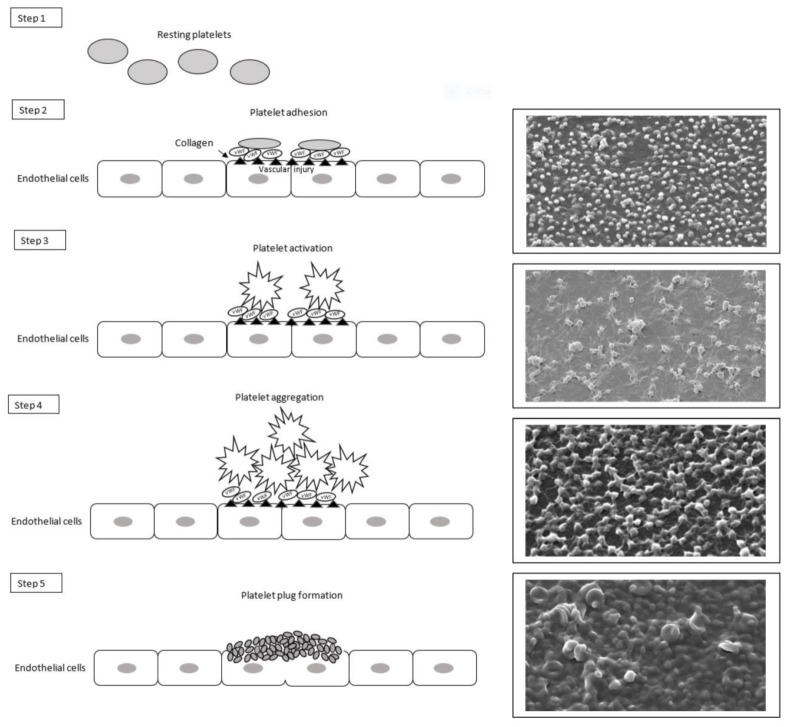
Scheme and scanning electron microscopy images of different steps of primary hemostasis. The resting platelets in the bloodstream (step 1) adhere to the endothelial or sub-endothelial matrix at site of the vascular lesion (step 2). Intra-platelet signal transduction induces platelet activation (step 3) followed by platelet aggregation (step 4) and platelet plug formation (step 5).

**Figure 3 life-11-01095-f003:**
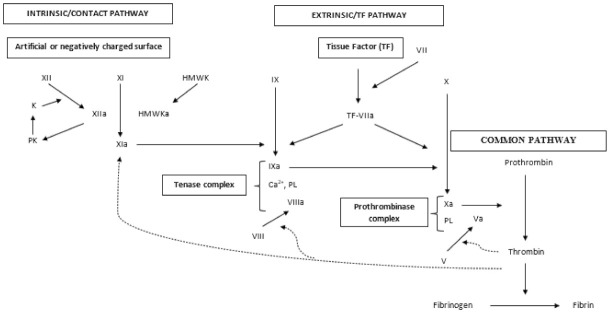
The coagulation cascade showing intrinsic, extrinsic, and common pathway, and feedback activation (dashed lines). PK = prekallikrein; K = kallikrein; HMWK = high molecular weight kininogen. PL = phospholipids.

**Table 1 life-11-01095-t001:** Plants cited in the text with metabolites present in the extracts and their antiplatelet/anticoagulant activity.

Family	Plants	Classes of Metabolites in the Extracts	Antiplatelet/Anticoagulant Activity of the Extracts
Oleaceae	*Olea europaea*	oleuropein, (+)-cycloolivil, hydroxytyrosol, hydroxytyrosol acetate,	in vitro inhibition of platelet aggregation and activation [120,121,122,123,124,125,126,127,128,129]
Asteracee	*Chamomilla recutita* L.	flavonoids apigenin, quercetin, patuletin, luteolin and their glucosides, terpenoids, alphabisabolol and its oxides, azulene, including chamazulene	in vitro inhibition of platelet aggregation [130,131,132,133]
Amaryllidaceae	*Allium sativum,* *Allium ursinum*	sulfur compounds (alliin, allicin, ajoene, allyl propyl disulfide, diallyl trisulfide (DATS), S-allylcysteine (SAC), vinyldithiins, S-allylmercaptocysteine), enzymes (al-liinase, peroxidases, tyrosinase), amino acids (arginine and others) and their glycosides, minerals	in vitro inhibition of platelet aggregation [134,135,136,137,138,139,140,141,142,143]
Lamiaceae	*Rosmarinus officinalis, Thymus atlanticus,* *Thymus zygis*	tritepenes, ursolic acid, oleanolic acid, betulinic acid, carnosol, micromeric acid, caffeic acid, rosmarinic acid, quercetin, rutin, hyperoside, luteolin-7-*O*-glucoside	in vitro prolongation of TT, in vivo inhibition of platelet aggregation [144,145,146,147,148,149,150,151], in vitro prolongation of APTT and PT [155,156,157,158,159,160,161,162,163]
Rosaceae and Asteraceae	*Fragaria vesca, Echinacea purpurea, Erigeron canadensis* L.	hexuronic acids and phenolic glycoconjugates	in vitro prolongation of APTT and PT [152,153,154]
Chrysobalanaceae	*Licania rigida*	gallic acid, catechin, chlorogenic acid, caffeic acid, epicatechin, ellagic acid, rutin, quercitrin, quercetin, kaempferol and kaempferol glycoside	in vitro prolongation of APTT and PT; anti-Xa and anti-IIa activity [164,165,166]
Papaveracee	*Fumaria officinalis*	phenolics and flavonoids	in vitro prolongation of APTT and PT [165,166]
Lecythidaceae	*Careya arborea*	3,4-dihydroxybenzoic acid, quercetin 3-O-glucopyranoside, kaempferol 3-Oglucopyranoside, quercetin 3-O-(6-O-glucopyranosyl)-gluco pyranoside, gallic acid,	in vitro prolongation of APTT, PT and TT [168,169,170,171,172]
Violaceae	*Viola yedoensis*	dimeresculetin	in vitro prolongation of APTT, PT and TT [173,174]
Euforbiacee	*Euphorbia resinifera*	serine protease EuRP-61	in vitro prolongation of APTT and PT and inhibition of platelet aggregation [175,176]
Orobanchaceae	*Orobanche caryophyllacea, Phelipanche arenaria, Phelipanche ramosa*	phenylpropanoid glycosides: tubuloside A, poliumoside, 3-*O*-methylpoliumoside	in vitro prolongation of APTT, PT and TT [177]
Rubiaceae	*Genipa americana*	glycoconjugates composed mainly by arabinose, galactose and uronic acid	in vitro prolongation of APTT and inhibition platelets aggregation; in vivo inhibition of venous thrombus formation and increasing of bleeding time [178,179]
Acanthaceae	*Pseuderanthemum palatiferum*	olyphenolic–polysaccharide conjugates	in vitro prolongation of APTT and PT [181,182]

## Data Availability

No new data were created or analyzed in this study. Data sharing is not applicable to this article.

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
