# Peer review of "Bioactive Natural Compounds with Antiplatelet and Anticoagulant Activity and Their Potential Role in the Treatment of Thrombotic Disorders"

_life, 2021, doi:10.3390/life11101095_

Round 1

Reviewer 1 Report

This is an exhaustive and interesting review on plant compounds with hemostatic effect.

It is important to indicate the concentration of the compound required to get functional consequences. For many cases, the concentration is too high, which considering the pleiotropic effects of these compounds, argue against a potential clinical usefulness for the treatments, as stated in the title of the manuscript. I suggest moderating the title and conclusion, and discussing this important limitation.

For most of compounds, only the final effect is known, but the underlying mechanism not. Please, indicate this fact.

I also miss a table with the main features of the most important compounds.

Finally, I also suggest including potential chemical modifications of natural compounds to improve their capacities (half-life, specificity, IC50…).

Minor points.

The pdf format I reviewed contains incomplete figures (Figure 1) or wrong formats (example in page 9).

Figure 2 might be improved.

Author Response

Referee 1

Thank you to the Referee for the useful comments and suggestions. The manuscript has been improved as suggested and the responses are listed below. The correction in the text have been reported in red.

  • It is important to indicate the concentration of the compound required to get functional consequences. For many cases, the concentration is too high, which considering the pleiotropic effects of these compounds, argue against a potential clinical usefulness for the treatments, as stated in the title of the manuscript. I suggest moderating the title and conclusion and discussing this important limitation.

The concentration of the compounds required to get functional consequences in patients is not known because mostly of the anticoagulant studies on plants extracts reported have been performed in vitro. Anyway, the concentration of extracts or purified molecules with anticoagulant and antiplatelet activity, when available in literature, have been added in the manuscript and the limitations in the use of plant extracts as anticoagulant drugs have been discussed (lines 572-577).

The title has been modified as suggested and not it is: “Bioactive natural compounds with antiplatelet and anticoagulant activity and their potential role in the treatment of thrombotic disorders”.

  • For most of compounds, only the final effect is known, but the underlying mechanism not. Please, indicate this fact.

The fact suggested has been reported in the text (lines 580-584).

  • I also miss a table with the main features of the most important compounds.

The table has been added (Table 1 pag. 15).

  • Finally, I also suggest including potential chemical modifications of natural compounds to improve their capacities (half-life, specificity, IC50…).

The potential chemical modifications of natural compounds to improve their capacities have been added (lines 585-593, reference n. 183).

Minor points.

  • The pdf format I reviewed contains incomplete figures (Figure 1) or wrong formats (example in page 9).

Figure 1 is now complete, and the incorrect formats have been corrected. However, in the uploaded file, the Figures were complete and there were no wrong formats in the text. So, I assume this inconvenient was due to the formatting procedure after uploading the manuscript online.

  • Figure 2 might be improved.

Figure 2 has been improved. Moreover, Scanning Electron Microscopy images of primary hemostasis steps have been added.

Reviewer 2 Report

Thank you for your research efforts.

Review articles on the treatment of thrombotic disorders and natural substances are impressive topics.

However, there are some errors in the written paper, so please correct it.

  1. The figure1. must fit on the designated page. Your figure cannot be checked beyond the page.
  2. Please delete the red under line on figure 2 and 3. Please attach clean version.
  3. Some paragraphs seem unorganized. Please arrange it in line with the other paragraphs.
  4. You have reviewed plants and their ingredients that are effective in thrombotic disorders. What was the criteria for selecting the plant? The answer to this question must be done.
  5. The conclusion part tends to be rather short compared to the contents of the review. Please add content.

Author Response

Referee 2

Thank you to the Referee for the useful comments and suggestions. The review has been improved as suggested and the responses are listed below. The correction in the text have been reported in red.

  • The figure1. must fit on the designated page. Your figure cannot be checked beyond the page.

Figure 1 has been modified as suggested.

  • Please delete the red under line on figure 2 and 3. Please attach clean version.
  • Some paragraphs seem unorganized. Please arrange it in line with the other paragraphs.

In the uploaded file, there were no wrong formats in the text. So, I assume this inconvenient was due to the formatting procedure after uploading the manuscript online. Anyway, a clean version of the manuscript has been attached.

  • You have reviewed plants and their ingredients that are effective in thrombotic disorders. What was the criteria for selecting the plant? The answer to this question must be done.

The main plants distributed in the world able to interfere with primary and secondary hemostasis were selected (lines 73-74).

  • The conclusion part tends to be rather short compared to the contents of the review. Please add content.

Content have been added to Conclusions.

Round 2

Reviewer 1 Report

The author has significantly improved her manuscript and has followed my suggestions.